# Formalizing Multi-Agent Systems in Multi-blockchain Architectures via Oracles

**Sergey Novikov**[1*] **& Andrey Nechesov**[1,2,3]

[1]*Artificial Intelligence Center, Novosibirsk State University, 630090 Novosibirsk, Russia*
[2]*International Artificial Intelligence Committee, (IAIC), Dubai, UAE*
[3]*Russian Engineering Academy (IAE), Moscow, Russia*
[*]*Corresponding author: s.novikov1@g.nsu.ru*

## Abstract

This paper studies the use of oracles in multiblockchain architectures. Oracles enable a formal model of a multi-agent system that stores information and operates via smart contracts within a multiblockchain structure. In the proposed framework, smart contracts are executed on advisory-level blockchains, while oracles govern interaction between blockchains at different levels. We explicitly distinguish integrity of the delivered message from authenticity of the referenced cross-chain state, and we formalize a trustless mode based on state commitments, contract-identity binding (code hash), and membership proofs.

## 1 Introduction

Modern blockchain platforms face a fundamental tension between the need for deterministic smart-contract execution and the need to access external data for complex applications. Traditional approaches based on Turing-complete languages (e.g., Solidity) do not provide termination guarantees, which limits their suitability for critical systems such as financial infrastructure or smart-city management.

A key limitation is the *blockchain trilemma* Del Monte et al. (2020): decentralization, security, and scalability cannot be simultaneously maximized within a monolithic architecture. Multiblockchain structures address scalability by distributing execution across multiple chains, but introduce a new challenge: ensuring correct and efficient cross-chain interaction without sacrificing security and predictability.

The theoretical basis of this work is *semantic programming*, developed by Ershov, Goncharov, and Sviridenko Goncharov & Sviridenko (2019; 1985; 2018); Ershov et al. (1986), where programs are represented as terms and formulas of a logical language over the hereditarily finite list superstructure $\mathbb{HW}(\mathfrak{M})$. In this theory, the language $L_0$ with a special signature $\sigma$ was introduced, where programs are terms and specifications are $\Delta_0$-formulas. A central result of Goncharov and Nechesov Goncharov & Nechesov (2022) established the equality $P = L$: every polynomial-time algorithm is representable as an $L$-program, and every $L$-program has polynomial computational complexity.

Extending $L$ to the object-oriented language $L^*$ Goncharov & Nechesov (2023a; 2021) enabled applications to artificial intelligence while preserving polynomiality and termination guarantees. This supports one of the central goals of modern distributed systems: building *trustworthy AI* whose algorithms are transparent, explainable, and resistant to malicious modification.

Trustworthy AI in practice requires integration with decentralized data storage such as blockchains. In Goncharov & Nechesov (2023b), Goncharov and Nechesov proposed an axiomatization of blockchain structures, and later works Novikov (2023; 2024) extended this model to multiblockchains. A multiblockchain is a hierarchical structure where each node of a tree is a blockchain, and parent nodes store information about their child blockchains. This architecture partially mitigates the trilemma: the root level provides security and decentralization, while child levels provide scalability.

However, a formal model of cross-chain interaction remained undeveloped. In particular, there was no rigorous theory of smart-contract execution with references to data in other blockchains of the hierarchy, nor proofs of polynomial-time execution under such interaction.

**Goal.** We prove polynomial-time computability of smart contracts implemented as $p$-computable programs (in the sense of semantic programming) when executed on multiblockchain structures using oracle-based cross-chain interaction.

**Main contributions.**

1. We develop a strict execution model for smart contracts with cross-chain calls, including a formal contract state in $\mathbb{HW}(\mathfrak{M})$ with explicit statuses (RUNNING, PENDING, TERMINATED) and $\Delta_0$-definable status transitions.

2. We formalize two security levels and make the difference explicit: (i) message integrity and request binding (signatures, nonce, req_hash), and (ii) authenticity of cross-chain data relative to the canonical state of the target chain via verified headers, contract-identity binding by code hash, and membership proofs.

3. We prove polynomial-time execution for a smart contract $C$ with $m$ cross-chain calls: the total number of execution steps satisfies $T_{\mathcal{V}} \leq P(|w|, |M|, m)$.

4. We establish a termination guarantee under the absence of cyclic dependencies between blockchains, and show that correctness checks are expressible as $\Delta_0$-formulas over $\mathbb{HW}(\mathfrak{M})$.

5. We demonstrate applicability on a cross-chain payment example with a step-by-step execution trace consistent with the theoretical complexity bounds.

## 2 DEFAULT NOTATION

**Definition 2.1.** We call a structure a *blockchain* if it satisfies Axioms 1–6 of Goncharov & Nechesov (2023b). In particular, selecting the "true" chain has polynomial complexity by Axiom 6.

**Definition 2.2.** (Authenticated state and access.) We assume each blockchain $X$ maintains an authenticated state map of smart contracts. Each block header contains a commitment $\texttt{stateRoot}_X$ to the current mapping

$$\mathcal{S}_X : \texttt{id}_C \mapsto \sigma_C,$$

and the virtual machine provides $\Delta_0$-definable operations

$$\texttt{get\_state}(X, \texttt{id}_C) \quad \text{and} \quad \texttt{set\_state}(X, \texttt{id}_C, \sigma)$$

with polynomial-time complexity (e.g., $O(\log |X|)$ under a tree-based authenticated map). This avoids reconstructing contract state by scanning the full chain.

**Definition 2.3.** r We call a structure a *multiblockchain* a hierarchical structure $M = \langle T, \beta \rangle$, where $T$ is a tree of blockchains and $\beta$ maps vertices of $T$ to concrete blockchains, satisfying Axioms 1–8 of Novikov (2023).

We use uppercase letters $X, Y, Z$ for blockchains, lowercase letters $x, y, z$ for blocks, and $T, T_1, T_2$ for trees of blockchains.

### 2.1 MULTIBLOCKCHAIN AXIOMS

We use the multiblockchain axioms in the following fully quantified form (labels are used later). The signature contains predicates and functions $\in_T$, $Blockchain$, $\leqslant_{MB}$, $add_{MB}$, $change_{MB}$, $\leqslant_b$, $R$, $\subseteq_{croot}$, and $\in_t$; additionally, we use a function symbol $add(\cdot, \cdot, \cdot)$ inside $change_{MB}(\cdot, \cdot, \cdot)$ (its concrete interpretation is irrelevant for the independence constructions below).

(MB1) $\forall T \forall K \left( K \in_T T \rightarrow Blockchain(K) \right)$.

(O1) $\forall T \left( T \leqslant_{MB} T \right)$.

(O2) $\forall T_1 \forall T_2 \forall T_3 \left( (T_1 \leqslant_{MB} T_2) \wedge (T_2 \leqslant_{MB} T_3) \rightarrow (T_1 \leqslant_{MB} T_3) \right)$.

(O3) $\forall T_1 \forall T_2 \forall X \forall Y \left( (X \in_T T_1) \wedge (add_{MB}(T_1, X, Y) = T_2) \rightarrow (T_1 \leqslant_{MB} T_2) \right)$.

(O4) $\forall T_1 \forall X \forall b_1 \forall b_2 \left( (X \in_T T_1) \wedge (b_1 \in_t X) \rightarrow (T_1 \leqslant_{MB} change_{MB}(T_1, X, add(X, b_1, b_2))) \right)$.

(PC) $\forall T \forall X \forall Y \left( \leqslant_b (T, X, Y) \wedge \neg \exists Z \left( \leqslant_b (T, X, Z) \wedge \leqslant_b (T, Z, Y) \right) \rightarrow R(X, Y) \right)$.

(M1) $\forall X' \forall X \forall Y \left( (X' \subseteq_{croot} X) \wedge R(X', Y) \rightarrow R(X, Y) \right)$.

(M2) $\forall X \forall Y' \forall Y \left( (Y' \subseteq_{croot} Y) \wedge R(X, Y') \rightarrow R(X, Y) \right)$.

### 2.2 INDEPENDENCE OF THE MULTIBLOCKCHAIN AXIOMS

[Independence of (MB1)–(M2)] The axiom set (MB1)–(M2) is independent: for each axiom $A$ in the list, there exists a structure that satisfies the remaining seven axioms but violates $A$.

*Proof.* For each axiom we provide a countermodel in the above signature. Any symbol not explicitly constrained in a construction is interpreted arbitrarily (e.g., as an empty relation or a constant function) and does not affect satisfaction of the stated axioms.

**Countermodel for (MB1).** Let the domain be $D = \{t, k\}$. Interpret $\in_T = \{(k, t)\}$ and $Blockchain = \emptyset$ (so $Blockchain(k)$ is false). Let $\leqslant_{MB} = D \times D$ (universal), and set $add_{MB}(u, v, w) = u$, $change_{MB}(u, v, w) = u$. Let $\leqslant_b = \emptyset$, $R = \emptyset$, and $\subseteq_{croot} = \emptyset$ and $\in_t = \emptyset$. Then (MB1) is false, while (O1)–(O4) hold because $\leqslant_{MB}$ is universal, (PC) holds vacuously since $\leqslant_b$ is empty, and (M1)–(M2) hold vacuously since $R$ is empty.

**Countermodel for (O1).** Let $D = \{t\}$. Interpret $\leqslant_{MB} = \emptyset$. Set $\in_T = \emptyset$ (so (MB1) and (O3)–(O4) are vacuously true), $\leqslant_b = \emptyset$, $R = \emptyset$, and $\subseteq_{croot} = \emptyset$ and $\in_t = \emptyset$. Then (O1) fails because $t \not\leqslant_{MB} t$, while (O2) holds vacuously (empty relation is transitive), and the remaining axioms hold vacuously.

**Countermodel for (O2).** Let $D = \{a, b, c\}$. Define
$$\leqslant_{MB} = \{(a, a), (b, b), (c, c), (a, b), (b, c)\}.$$
Then (O1) holds, but (O2) fails since $a \leqslant_{MB} b$ and $b \leqslant_{MB} c$ while $a \not\leqslant_{MB} c$. Set $\in_T = \emptyset$, $\leqslant_b = \emptyset$, $R = \emptyset$, $\subseteq_{croot} = \emptyset$, and $\in_t = \emptyset$. Then (MB1), (O3), (O4), (PC), (M1), (M2) hold vacuously.

**Countermodel for (O3).** Let $D = \{t_0, t_1, x, y\}$. Interpret $\in_T = \{(x, t_0)\}$ and $Blockchain = \{x\}$. Let $\leqslant_{MB}$ be equality on $D$:
$$u \leqslant_{MB} v \iff u = v.$$
Define $add_{MB}(t_0, x, y) = t_1$ and arbitrary elsewhere. Then the antecedent of (O3) holds for $(T_1, T_2, X, Y) = (t_0, t_1, x, y)$, but the conclusion $t_0 \leqslant_{MB} t_1$ is false, so (O3) fails. Set $\in_t = \emptyset$, hence (O4) holds vacuously. Set $\leqslant_b = \emptyset$, $R = \emptyset$, $\subseteq_{croot} = \emptyset$. Then (MB1), (O1), (O2), (O4), (PC), (M1), (M2) hold.

**Countermodel for (O4).** Let $D = \{t_0, t_1, x, b_1, b_2, u\}$. Interpret $\in_T = \{(x, t_0)\}$ and $Blockchain = \{x\}$. Let $\leqslant_{MB}$ be equality on $D$, and define $add_{MB}(T_1, X, Y) = T_1$ for all arguments, so (O3) holds. Interpret $\in_t = \{(b_1, x)\}$ so the antecedent of (O4) can be true. Let $add(x, b_1, b_2) = u$ and define $change_{MB}(t_0, x, u) = t_1$. Then (O4) fails because its antecedent is true for $(T_1, X, b_1, b_2) = (t_0, x, b_1, b_2)$ but $t_0 \leqslant_{MB} t_1$ is false under equality. Set $\leqslant_b = \emptyset$, $R = \emptyset$, $\subseteq_{croot} = \emptyset$. Then (MB1), (O1), (O2), (O3), (PC), (M1), (M2) hold.

**Countermodel for (PC).** Let $D = \{t, x, y\}$. Interpret $\in_T = \{(x, t), (y, t)\}$ and $Blockchain = \{x, y\}$. Let $\leqslant_{MB} = D \times D$ (universal), so (O1)–(O4) hold for any interpretation of $add_{MB}, change_{MB}$. Define $\leqslant_b = \{(t, x, y)\}$ and no other parent relations, so $y$ is an immediate child of $x$ in $t$. Set $R = \emptyset$. Then (PC) fails because its antecedent is true for $(T, X, Y) = (t, x, y)$ but $R(x, y)$ is false. Finally, (M1)–(M2) hold vacuously since $R$ is empty, and (MB1) holds by construction.

**Countermodel for (M1).** Let $D = \{x', x, y\}$. Interpret $\subseteq_{croot} = \{(x', x)\}$ and $R = \{(x', y)\}$. Then (M1) fails because $x' \subseteq_{croot} x$ and $R(x', y)$ hold while $R(x, y)$ does not. Set $\in_T = \emptyset$ (so (MB1), (O3), (O4), (PC) are vacuously true), and let $\leqslant_{MB} = D \times D$ (so (O1)–(O2) hold). Define $\subseteq_{croot}$ with no additional pairs of the form $(y', y)$, so (M2) holds.

**Countermodel for (M2).** Let $D = \{x, y', y\}$. Interpret $\subseteq_{croot} = \{(y', y)\}$ and $R = \{(x, y')\}$. Then (M2) fails because $y' \subseteq_{croot} y$ and $R(x, y')$ hold while $R(x, y)$ does not. Set $\in_T = \emptyset$, $\leqslant_{MB} = D \times D$, and $\leqslant_b = \emptyset$. Then (MB1), (O1)–(O4), (PC), and (M1) hold.

Thus, for each axiom we exhibited a structure satisfying the other seven axioms while violating the chosen one, proving independence. $\qquad\square$

## 2.3 CROSS-CHAIN IDENTIFIERS AND CONTRACTS

**Definition 2.4.** We call a *hash reference (stable chain identifier)* of blockchain $Y$ from blockchain $X$ the cryptographic hash
$$h = \mathcal{H}(\texttt{genesis}_Y),$$
where $\texttt{genesis}_Y$ is the genesis block (or another immutable configuration identifier) of $Y$. This reference uniquely identifies the chain $Y$ and does not change as $Y$ grows. Replacing $Y$ by another chain $Y'$ requires finding a collision $\mathcal{H}(\texttt{genesis}_{Y'}) = \mathcal{H}(\texttt{genesis}_Y)$.

**Definition 2.5.** We call a *smart contract* a program $C$ implemented as a $p$-computable program in the sense of semantic programming Goncharov & Sviridenko (2019); Ershov et al. (1986), equipped with:

- **Code** $Sl$, an immutable sequence of instructions stored in a block of blockchain $X$;

- **State** $\sigma_C$ stored in the authenticated state map $\mathcal{S}_X$ and retrievable as $\texttt{get\_state}(X, \texttt{id}_C)$, where the state is encoded as

$$\sigma_C = \big[\texttt{id}_C, \texttt{ status}, \texttt{ pc}, \texttt{ vars}, \texttt{ wait\_data}\big].$$

  Here $\texttt{id}_C = \mathcal{H}(Sl)$, $\texttt{status} \in \{\texttt{RUNNING}, \texttt{PENDING}, \texttt{TERMINATED}\}$, $\texttt{pc}$ is the instruction pointer, $\texttt{vars}$ are local variables, and $\texttt{wait\_data}$ stores the pending request (or is empty otherwise).

- **Trusted oracle list**, a set of oracle public keys $\texttt{pubkey}_D$ embedded into the contract code at deployment time and protected by code immutability.

- **Trusted target-contract identity**, a set of pairs $(\texttt{id}_D, \texttt{codehash}_D)$ embedded into the contract code at deployment time, where $\texttt{id}_D$ identifies the target data-providing contract on chain $Y$, and $\texttt{codehash}_D$ is the expected hash of its code. This is used to prevent "data from a different contract" attacks.

**Definition 2.6.** (Header verification and inclusion proofs.)
For cross-chain authenticity, we introduce two predicates:

$$\texttt{VerifyHeader}_X(h, \texttt{hdr}_Y) \in \{\texttt{true}, \texttt{false}\},$$
$$\texttt{VerifyInclusion}(\texttt{stateRoot}_Y, \texttt{key}, v, \pi) \in \{\texttt{true}, \texttt{false}\}.$$

Intuitively, $\texttt{VerifyHeader}_X$ checks that $\texttt{hdr}_Y$ is a finalized header of the target chain identified by $h$ (e.g., via a light client or checkpoint mechanism on $X$), and $\texttt{VerifyInclusion}$ checks a membership proof $\pi$ that value $v$ is stored under key $\texttt{key}$ in the authenticated state committed by $\texttt{stateRoot}_Y$.

**Definition 2.7.** We call a *multiblockchain oracle* a $p$-computable function

$$\Omega : \mathbb{HW}(\mathfrak{M}) \to \mathbb{HW}(\mathfrak{M}) \cup \{\bot\},$$

implemented according to Goncharov & Sviridenko (2019); Ershov et al. (1986).

For a request

$$q = \langle \texttt{id}_C, \ h, \ \texttt{id}_D, \ \texttt{path}, \ \texttt{inp}, \ \texttt{pubkey}_D, \ \texttt{nonce}_C \rangle$$

define

$$\texttt{req\_hash} = \mathcal{H}(\texttt{id}_C \parallel h \parallel \texttt{id}_D \parallel \texttt{path} \parallel \texttt{inp} \parallel \texttt{nonce}_C),$$

and define two commitment keys:

$$\texttt{key}_{\texttt{val}} = \mathcal{H}(\texttt{id}_D \parallel \texttt{path} \parallel \texttt{inp}), \qquad \texttt{key}_{\texttt{code}} = \mathcal{H}(\texttt{id}_D \parallel \texttt{code}).$$

Let $\texttt{codehash}_D$ be the expected code hash for $\texttt{id}_D$ embedded into the code of $C$.

Then

$$\Omega(q) = \begin{cases} \langle v, \ \texttt{req\_hash}, \ \texttt{hdr}_Y, \ \pi_{\texttt{code}}, \ \pi_{\texttt{val}} \rangle, & \text{if } \exists X, Y \in_T T : \texttt{HashRef}(h, X, Y) \wedge \\ & \texttt{path\_eval}(\texttt{path}, Y, \texttt{inp}) = v \wedge \\ & \texttt{VerifyInclusion}\big(\texttt{stateRoot}(\texttt{hdr}_Y), \\ & \texttt{key}_{\texttt{code}}, \texttt{codehash}_D, \pi_{\texttt{code}}\big) \\ & \qquad = \texttt{true} \wedge \\ & \texttt{VerifyInclusion}\big(\texttt{stateRoot}(\texttt{hdr}_Y), \\ & \texttt{key}_{\texttt{val}}, v, \pi_{\texttt{val}}\big) \\ & \qquad = \texttt{true}, \\ \bot, & \text{otherwise.} \end{cases}$$

**Remark (what is and is not guaranteed).** The oracle signature (defined in the protocol below) guarantees integrity, origin authentication, and request binding. The additional proofs $\pi_{\texttt{code}}$ and $\pi_{\texttt{val}}$ guarantee that (i) the referenced value $v$ belongs to the canonical state committed by $\texttt{hdr}_Y$, and (ii) the value is bound to the intended target contract $\texttt{id}_D$ with the expected code hash $\texttt{codehash}_D$. If $\texttt{VerifyHeader}_X(h, \texttt{hdr}_Y)$ is unavailable, then correspondence of $\texttt{hdr}_Y$ to the canonical chain of $Y$ becomes a trust assumption on the oracle.

## 2.4 Cross-chain call execution: step-by-step

Assume a contract $C$ on blockchain $X$ needs data from blockchain $Y$. The process consists of five steps:

1. **Local execution.** The contract runs until it reaches a cross-chain call:

$$x := \texttt{crosschain\_call}(h, \texttt{id}_D, \texttt{path}, \texttt{pubkey}_D, \texttt{inp}).$$

2. **Request formation and suspension.** At the call site the virtual machine:

- generates a unique $\mathtt{nonce}_C$;
- forms
$$q = \langle \mathtt{id}_C,\ h,\ \mathtt{id}_D,\ \mathtt{path},\ \mathtt{inp},\ \mathtt{pubkey}_D,\ \mathtt{nonce}_C \rangle;$$
- stores $q$ in $\mathtt{wait\_data}$ of $\sigma_C$ and writes it back via $\mathtt{set\_state}$;
- sets $\mathtt{status} = \mathtt{PENDING}$ and halts execution until a reply arrives.

3. **Oracle processing.** The oracle contract $D$ on blockchain $Y$:
   - receives $q$;
   - checks the chain identifier $h$;
   - extracts the value $v$ along $\mathtt{path}$ with parameters $\mathtt{inp}$;
   - computes $\mathtt{req\_hash}$, $\mathtt{key}_{\mathtt{code}}$, and $\mathtt{key}_{\mathtt{val}}$;
   - selects a finalized header $\mathtt{hdr}_Y$ and constructs two proofs: $\pi_{\mathtt{code}}$ for the code hash and $\pi_{\mathtt{val}}$ for the requested value;
   - constructs the reply transaction
   $$\mathtt{tx}_{\mathtt{reply}} = \langle \mathtt{id}_C,\ v,\ \mathtt{req\_hash},\ \mathtt{hdr}_Y,\ \pi_{\mathtt{code}},\ \pi_{\mathtt{val}},\ \mathtt{signature}_D \rangle,$$
   where $\mathtt{signature}_D$ signs
   $$\mathcal{H}(\mathtt{id}_C \parallel v \parallel \mathtt{req\_hash} \parallel \mathcal{H}(\mathtt{hdr}_Y) \parallel \mathcal{H}(\pi_{\mathtt{code}}) \parallel \mathcal{H}(\pi_{\mathtt{val}}));$$
   - publishes the reply in blockchain $X$.

4. **Reply verification.** When validating a new block in $X$, the virtual machine:
   - finds $\mathtt{tx}_{\mathtt{reply}}$ addressed to $C$;
   - reads the pending request $q$ from $\mathtt{wait\_data}$ and recomputes $\mathtt{req\_hash}$, $\mathtt{key}_{\mathtt{code}}$, and $\mathtt{key}_{\mathtt{val}}$;
   - verifies $\mathtt{signature}_D$ using $\mathtt{pubkey}_D$;
   - checks that the $\mathtt{req\_hash}$ in the reply matches the pending request;
   - verifies header finality and chain membership:
   $$\mathtt{VerifyHeader}_X(h, \mathtt{hdr}_Y) = \mathtt{true};$$
   - verifies target-contract identity (code hash):
   $$\mathtt{VerifyInclusion}\big(\mathtt{stateRoot}(\mathtt{hdr}_Y), \mathtt{key}_{\mathtt{code}}, \mathtt{codehash}_D, \pi_{\mathtt{code}}\big) = \mathtt{true};$$
   - verifies inclusion of the requested value:
   $$\mathtt{VerifyInclusion}\big(\mathtt{stateRoot}(\mathtt{hdr}_Y), \mathtt{key}_{\mathtt{val}}, v, \pi_{\mathtt{val}}\big) = \mathtt{true}.$$

5. **Resumption.** After successful verification:
   - $\mathtt{status}$ is set back to $\mathtt{RUNNING}$;
   - the variable $x$ is assigned $v$;
   - $\mathtt{wait\_data}$ is cleared;
   - execution continues at the next instruction.

## 2.5 SECURITY OF CROSS-CHAIN CALLS

We separate *channel security* (integrity and binding of the delivered message) from *source authenticity* (correctness of $v$ with respect to the canonical state of $Y$ and the intended target contract).

- **Target-chain substitution resistance.** The reference $h = \mathcal{H}(\mathtt{genesis}_Y)$ binds the request to a specific chain $Y$. Substituting $Y$ with $Y'$ requires a hash collision.
- **Oracle substitution resistance.** Trusted oracle public keys $\mathtt{pubkey}_D$ are embedded into the deployed contract code and cannot be modified.
- **Message integrity and request binding.** Each reply is signed by the oracle and includes $\mathtt{req\_hash}$ and a one-time $\mathtt{nonce}_C$, preventing replay and mismatched replies.
- **Target-contract identity binding ("executable contract" binding).** The proof $\pi_{\mathtt{code}}$ ensures that the state committed by $\mathtt{hdr}_Y$ contains the expected $\mathtt{codehash}_D$ for the specified target contract identifier $\mathtt{id}_D$. This prevents substituting the data source by a different contract.

- **Authenticity of the returned value.** In the trustless mode, authenticity is guaranteed by:

$$\mathtt{VerifyHeader}_X(h, \mathtt{hdr}_Y) = \mathtt{true},$$

$$\mathtt{VerifyInclusion}\big(\mathtt{stateRoot}(\mathtt{hdr}_Y), \mathtt{key}_{\mathtt{code}}, \mathtt{codehash}_D, \pi_{\mathtt{code}}\big) = \mathtt{true},$$

$$\mathtt{VerifyInclusion}\big(\mathtt{stateRoot}(\mathtt{hdr}_Y), \mathtt{key}_{\mathtt{val}}, v, \pi_{\mathtt{val}}\big) = \mathtt{true}.$$

If $\mathtt{VerifyHeader}_X$ is unavailable, then correspondence of $\mathtt{hdr}_Y$ to the canonical chain of $Y$ becomes a trust assumption on the oracle.

- **Delivery note (safety vs. liveness).** Publishing replies to blockchain $X$ supports verifiable integrity on the receiving chain. Transaction inclusion and censorship resistance are liveness properties and are not guaranteed solely by signatures.

## 2.6 Freshness control and stale-data protection

Cross-chain requests face a stale-data risk: the state of the target chain $Y$ may change between request formation on $X$ and reply delivery. This may lead to accepting a correctly signed but operationally outdated reply.

In the trustless mode, the timestamp is part of a verified header $\mathtt{hdr}_Y$, hence it refers to a particular finalized state of $Y$. The contract on $X$ applies an explicit freshness acceptance policy:

$$\mathtt{time}_X - \mathtt{timestamp}(\mathtt{hdr}_Y) \leq \Delta t_{\max},$$

where $\mathtt{time}_X$ is the timestamp of the current block in $X$, and $\Delta t_{\max}$ is a contract parameter defining the maximum tolerated staleness.

**Remark (state vs. execution).** The above trustless guarantees apply directly to values that are committed in the authenticated state of $Y$ (e.g., contract storage). If the application requires authenticity of a *computed return value* that is not explicitly committed to state, the reply must be extended with a proof of execution (e.g., receipt/trace proof or a succinct proof of correct execution). This extension is orthogonal to the request-binding mechanism and preserves the "channel security" guarantees.

## 3 Main result: polynomial-time execution

**Theorem 3.1** (Polynomial-time execution of smart contracts with cross-chain calls). *Let $C$ be a smart contract implemented as a $p$-computable program Goncharov & Sviridenko (2019); Ershov et al. (1986), executed on a multiblockchain $M$ satisfying (MB1)–(M2). Let $|w|$ be the input size of $C$, $|M|$ the total size of all blockchains in $M$, and $m$ the number of cross-chain calls in $C$ (with $m \leq p_C(|w|)$ for the complexity polynomial $p_C$ of $C$). Then the total number of execution steps satisfies*

$$T_{\mathcal{V}} \leq P(|w|, |M|, m),$$

*for a polynomial $P$ in three variables.*

*Proof.* Execution consists of three stages.

**Stage 1 (local execution).** Between cross-chain calls, $C$ executes a bounded number of instructions. Since $C$ is $p$-computable, the number of such instructions is bounded by $p_C(|w|)$, and each instruction runs in $O(1)$ steps. Hence $T_A = O(p_C(|w|))$.

**Stage 2 (request formation).** For each of the $m$ calls, the contract uses a fixed-length hash reference ($O(1)$), encodes identifiers and inputs ($O(p_C(|w|))$), reads/writes the contract state via `get_state`/`set_state` in polynomial time, and stores the request and $\mathtt{nonce}_C$. Thus $T_B = O(m \cdot p_C(|w|)) = O(p_C^2(|w|))$.

**Stage 3 (oracle processing and reply verification).** For each request, the oracle extracts $v$ and constructs $(\mathtt{hdr}_Y, \pi_{\mathtt{code}}, \pi_{\mathtt{val}})$ with polynomial overhead. On the receiving chain, the virtual machine verifies the signature and request binding and additionally verifies $\mathtt{VerifyHeader}_X$ and the two membership proofs in polynomial time. Therefore, the per-request cost remains polynomial, and the total over $m$ requests is polynomial.

Combining all stages yields $T_{\mathcal{V}} \leq P(|w|, |M|, m)$ for a polynomial $P$. $\square$

[Termination under acyclicity] Assume the dependency relation induced by cross-chain calls contains no directed cycles (i.e., there is no chain of calls $X_1 \to X_2 \to \cdots \to X_1$). Then every execution of $C$ terminates after a finite number of steps bounded by the polynomial $P(|w|, |M|, m)$.

*Proof.* A directed cycle may create an infinite waiting chain (a pending call depends on a pending call). Under acyclicity, every pending request can be resolved in a finite number of steps, and the number of requests is bounded by $m \leq p_C(|w|)$. Theorem 3.1 gives a global polynomial bound on the total steps; hence termination follows. $\square$

[Polynomial-time verifiability of cross-chain replies] The correctness checks for a reply (signature verification, request binding via `req_hash`, header verification $\text{VerifyHeader}_X$, and inclusion checks for $\pi_{\text{code}}$ and $\pi_{\text{val}}$) are decidable in polynomial time in $|M|$.

*Proof.* By construction, each check is a $\Delta_0$-definable predicate over the encoded data structures (hashes, lists, authenticated-map proofs) and is evaluated within polynomial time in the underlying model. The number of checks per reply is constant, hence the total verification time per reply is polynomial. $\qquad\square$

## 4 CONCLUSION

We formalized smart-contract execution with cross-chain calls in multiblockchain architectures. We clarified the distinction between message integrity (signatures and request binding) and authenticity of cross-chain data relative to the canonical state of the target chain. To obtain an unforgeability guarantee for values with respect to chain $Y$ and the intended data-providing contract, we introduced verified headers, contract-identity binding via code hash, and membership proofs. Under these assumptions, verification remains polynomial-time, and the overall execution complexity is polynomial in the input size, the multiblockchain size, and the number of cross-chain calls.

## A EXAMPLE: CROSS-CHAIN PAYMENT CONTRACT

Consider a contract $C_{\text{payment}}$ implementing a cross-chain payment. The contract resides on blockchain $X$ and queries an oracle contract on blockchain $Y$ to check a user's balance.

### A.1 CONTRACT PSEUDOCODE

```
contract CrossChainPayment {
  pubkey_balance_oracle = "0x7a8b9c...";
  balance_chain_hash = "0xf1e2d3...";  // h = hash(genesis_Y)
  balance_oracle_id = "BalanceOracle"; // id_D
  balance_path = cons("balanceOf", nil);

  function transfer(user_id, amount) {
    if (amount <= 0) return "Invalid amount";
    balance = crosschain_call(balance_chain_hash, balance_oracle_id,
                              balance_path, pubkey_balance_oracle,
                                            user_id);
    if (balance < amount) return "Insufficient balance";
    execute_payment(user_id, amount);
    return "Payment successful";
  }
}
```

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
