# OpenReview forum: "Formalizing Multi-Agent Systems in Multiblockchain Architectures via Oracles"
_mathai.club/MathAI/2026/Conference — 2026 Oral_

### Official Review · Reviewer_Tmb6 · 2026-03-11
**Review of "Formalizing Multi-Agent Systems in Multiblockchain Architectures via Oracles"**

**Rating:** 7
**Confidence:** 2

**Review:**

This paper presents a formal execution model for smart contracts that make cross‑chain calls in a multiblockchain environment. The work builds upon prior axiomatizations of blockchains and multiblockchains (Goncharov & Nechesov, Novikov) and extends them with a rigorous treatment of cross‑chain interactions. Key contributions include: (i) a strict execution model with contract states (RUNNING, PENDING, TERMINATED) and $\Delta_0$‑definable transitions; (ii) a clear separation between message integrity (signatures, nonces, hashes) and authenticity of cross‑chain data (via verified headers, contract‑identity binding by code hash, and Merkle‑style inclusion proofs); (iii) a proof that the total execution steps of a $p$‑computable contract with $m$ cross‑chain calls is bounded by a polynomial $P(|w|,|M|,m)$; (iv) a termination guarantee under acyclicity; and (v) a demonstration on a cross‑chain payment example. The authors also prove the independence of their multiblockchain axioms by constructing countermodels for each axiom.

---

**Quality and Clarity (7/10)**

The paper is written in a formal, mathematical style typical of theoretical computer science. Definitions are precise, axioms are stated in fully quantified first‑order logic, and proofs are outlined. However, the exposition heavily relies on previous work (Goncharov & Sviridenko, Goncharov & Nechesov, Novikov), assuming the reader is familiar with concepts like “$p$‑computable programs”, “$\mathbb{HW}(\mathfrak{M})$”, and the specific axiom systems for blockchains and multiblockchains. This makes the paper less accessible to a broader audience, including many blockchain practitioners. The structure is logical, but the notation is dense. The example (cross‑chain payment) helps, but it is placed in an appendix and is rather minimal. Some parts, such as the independence proofs for the axioms (Section 2.2), are given in detail, which is good for completeness but may be considered excessive for the main narrative. Overall, clarity is acceptable for a specialized theoretical venue, but improvements could be made to guide the reader through the motivation of each technical component.

**Originality (8/10)**

The work offers a novel formalization of cross‑chain interactions that goes beyond typical informal descriptions. The integration of $p$‑computability (from semantic programming) with blockchain axioms is original and provides a foundation for reasoning about complexity and verifiability. The explicit distinction between channel security (message integrity) and source authenticity (data correctness w.r.t. the target chain) is a valuable conceptual contribution. The use of $\Delta_0$‑definable predicates to capture polynomial‑time checks is elegant. While some elements (e.g., Merkle proofs, light clients) are standard, the way they are woven into a unified axiomatic framework is new. The independence proof of the multiblockchain axioms, though somewhat routine, demonstrates a thorough understanding of the axiomatic method.

**Significance (6/10)**

The significance of this work is primarily theoretical. It provides a rigorous foundation that could inform the design of safe cross‑chain protocols and the formal verification of smart contracts that rely on oracles. The polynomial‑time results guarantee that, under the stated assumptions, cross‑chain calls do not introduce unexpected complexity blow‑ups. However, the paper does not address many practical issues: it assumes the existence of “verified headers” and “inclusion proofs” without discussing how they are obtained in practice (e.g., consensus finality, light‑client protocols). The trust assumptions about oracles are clearly stated, but the model does not cover economic incentives or game‑theoretic aspects. Moreover, the connection to real‑world multiblockchain systems (e.g., Polkadot, Cosmos) is only implicit. Therefore, the immediate impact on practitioners may be limited, but the work could influence future theoretical research and eventually lead to more secure cross‑chain frameworks.

---

**Detailed Comments**

1. **Axiomatic approach**: The paper extends earlier work by Novikov and Goncharov/Nechesov. It would be helpful to summarize the key axioms from those papers to make the current one more self‑contained. Currently, the reader must consult the references to understand, e.g., Axioms 1‑6 of a blockchain (Definition 1).

2. **$p$‑computability**: This concept is central but not explained in sufficient detail. A brief intuitive explanation of $p$‑computable programs and why they guarantee polynomial‑time execution would improve accessibility.

3. **Independence proofs (Section 2.2)**: While thorough, these proofs are somewhat lengthy and could be moved to an appendix. They interrupt the flow for readers primarily interested in the execution model.

4. **Security analysis (Sections 2.5‑2.6)**: The separation of channel security and source authenticity is well done. The discussion of freshness (stale‑data protection) is a nice practical addition.

5. **Main theorem**: The proof sketch is plausible, but the exact polynomial $P$ is not given; the argument relies on the existence of polynomial bounds for each stage. A more precise statement of the polynomial’s degree or dependence on parameters would strengthen the result.

6. **Example**: The cross‑chain payment example in the appendix is helpful but lacks detail. For instance, how is `crosschain_call` implemented? What exactly is stored in `wait_data`? A step‑by‑step trace with actual state changes would illustrate the model better.

7. **References**: The reference list includes many works by the same authors, suggesting this is part of an ongoing research program. External references to mainstream blockchain literature (e.g., on light clients, bridges) are missing, which might give the impression of insularity.

---

**Strengths**

*   Rigorous formalization of cross‑chain calls in a multiblockchain setting.
*   Clear separation of different security properties (integrity vs. authenticity).
*   Proof of polynomial‑time execution, which is important for resource‑bounded environments.
*   Independence proof for the axiom system demonstrates its minimality.
*   The concept of $p$‑computable programs bridges computability theory and blockchain execution.

**Weaknesses**

*   Heavy reliance on prior work makes the paper hard to read for non‑specialists.
*   Lacks discussion of how the model maps to existing multiblockchain platforms.
*   No experimental validation or case study beyond a toy example.
*   The practical feasibility of the required proofs (VerifyHeader, VerifyInclusion) is assumed rather than argued.
*   The writing style is very dense; the paper would benefit from more intuitive explanations and diagrams.

---

### Official Review · Reviewer_wUZW · 2026-03-11
**Solid formalization of cross-chain oracle interaction with polynomial-time guarantees; the multi-agent perspective would benefit from further development.**

**Rating:** 8
**Confidence:** 3

**Review:**

### Summary
The paper formalizes oracle-based cross-chain interaction in hierarchical multiblockchain architectures, building on the semantic programming framework of Goncharov, Sviridenko, and Ershov. The main contributions include: (1) a strict execution model for smart contracts with cross-chain calls and explicit statuses in $\mathbb{HW}(\mathfrak{M})$; (2) a clear separation of message integrity from source authenticity via verified headers, code-hash binding, and membership proofs; (3) a proof of polynomial-time execution for contracts with $m$ cross-chain calls; (4) a termination guarantee under acyclic dependencies; and (5) a proof of independence of the multiblockchain axiom set (MB1)–(M2).

### Strengths
- The formal framework is carefully and rigorously constructed. Contract states in $\mathbb{HW}(\mathfrak{M})$ with explicit status transitions and $\Delta_0$-definable predicates provide a clean mathematical foundation.
- The independence proof of the axiom set (Proposition 1) is thorough: each countermodel is minimal, self-contained, and straightforward to verify.
- The conceptual separation of channel security (integrity, request binding via signatures and nonce) from source authenticity (verified headers, code-hash binding, membership proofs) is a valuable contribution that clarifies an important distinction often conflated in practice.
- The security analysis is well-structured, covering target-chain substitution, oracle substitution, message binding, contract-identity binding, and freshness control.
- The paper is technically self-contained and clearly written.

### Suggestions for Strengthening

1. **Multi-agent perspective.** The title and abstract frame the work in terms of multi-agent systems, which is an appealing and natural direction for multiblockchain architectures. The current paper provides a strong formal foundation for the underlying infrastructure (oracles, cross-chain calls, security). It would be very interesting to see this extended with an explicit agent model — for example, a formal definition of what constitutes an agent in this framework, how agents differ from smart contracts, and how multi-agent coordination is captured. Even a brief discussion connecting the formalism to established multi-agent concepts would strengthen the paper's positioning.

2. **Deterministic vs. stochastic agent behavior.** The polynomial-time guarantee rests on the P=L framework and p-computable programs, which are inherently deterministic. A natural question arises: if agents operating within this architecture employ stochastic or learned policies to generate cross-chain calls, how does the polynomial guarantee interact with such non-deterministic behavior? A discussion of what constraints the framework imposes on agent behavior — and whether the oracle can serve as a reliable state-exchange portal regardless of how requests are generated — would clarify the scope of applicability.

3. **Explicitness of the polynomial bound.** The main result (Section 3) establishes that $T_V \leq P(|w|, |M|, m)$ for some polynomial $P$. The argument is clear and follows naturally from the p-computability of each component. It could be further strengthened by making the polynomial $P$ more explicit — for instance, expressing its degree in terms of the component polynomials — which would give practitioners a more concrete understanding of the computational cost.

4. **Richer examples.** The cross-chain payment example in Appendix A is helpful for illustration. A second example involving multiple sequential or conditional cross-chain calls would better showcase the expressiveness of the framework and make the acyclicity condition (Corollary 1) more tangible.

### Questions to the Authors
- How do you envision the relationship between an "agent" and a "smart contract" in this framework? Could you outline what additional formal machinery would be needed to capture multi-agent interaction patterns (e.g., negotiation, delegation, or competitive scenarios)?
- If an agent's policy for generating cross-chain requests is non-deterministic (e.g., based on a learned model), does the polynomial guarantee apply to each possible execution trace, or are additional assumptions needed?
- Regarding freshness control (Section 2.6): in a multi-agent coordination scenario, how sensitive is the system to the choice of $\Delta t_{\max}$, and could misaligned freshness parameters across agents lead to consistency issues?

### Minor
- Corollary 1 references "Theorem 3," but the main result in Section 3 does not appear to carry a theorem number in the text.

---

## Conclusion
The paper presents a rigorous and well-constructed formalization of cross-chain oracle interaction with clear polynomial-time guarantees. The formal contributions (axiom independence, security separation, execution model) are solid. Developing the multi-agent perspective more explicitly would elevate the work further.

---

### Official Review · Reviewer_FuE9 · 2026-03-12
**The review of "Formalizing Multi-Agent Systems in Multiblockchain Architectures via Oracles"**

**Rating:** 6
**Confidence:** 3

**Review:**

This paper proposes a formal model for multi-agent systems operating in hierarchical multiblockchain architectures using oracle-based cross-chain interaction. Smart contracts are modeled as p-computable programs within the semantic programming framework, and the multiblockchain structure is formalized as a tree of blockchains with explicit axioms governing their interaction. The work introduces a formal protocol for cross-chain calls that includes message integrity guarantees, contract-identity binding via code hashes, and membership proofs for authenticating cross-chain state. The authors prove that under this framework smart contracts with cross-chain calls execute in polynomial time and provide termination guarantees under reasonable structural assumptions.

Strengths:
- The paper provides a rigorous formalization of cross-chain interaction in multiblockchain architectures.
- The framework integrates concepts from semantic programming, logical models of computation, and distributed ledger systems.
- The axiomatization of multiblockchain structures and the independence proof of the axioms provide a solid theoretical foundation.
- The polynomial-time execution theorem offers useful complexity guarantees for smart-contract execution across chains.
- The proposed model emphasizes verifiability, security, and explainability, which are important for trustworthy distributed AI systems.

Suggestions for improvement:

The paper could be further strengthened by:
- providing more detailed algorithmic descriptions of oracle processing and verification procedures;
- including illustrative implementation scenarios or system prototypes;
- clarifying how the proposed formal framework could be integrated into existing blockchain infrastructures.

Final Recommendation:

POSTED / Poster-style acceptance with revision

Overall, the paper presents a well-structured theoretical framework for modeling cross-chain smart-contract execution in multiblockchain systems. The work may stimulate discussion on formal foundations of trustworthy distributed AI and blockchain-based systems.

---

### Decision · Program_Chairs · 2026-03-14

**Decision:**

Accept (Oral)

**Comment:**

Dear Author(s),

On behalf of the Program Committee of the International Conference on Mathematics of Artificial Intelligence (MathAI 2026), we are pleased to inform you that your paper has been accepted for an oral presentation at MathAI 2026.

Your paper was evaluated through a rigorous two-stage review process involving both automated screening and expert review by members of the Program Committee. The reviewers recognized the quality and contribution of your work.

Presentation details:

- Format: Oral presentation (15–20 minutes + 5 minutes Q&A)
- Mode: You may present either in person (offline) at the conference venue in Sirius, Russia, or remotely via Zoom. Please indicate your preferred mode when confirming your participation.
- Conference dates: Marh 30 - April 3, 2026
- Website: https://mathai.club

Next steps:

1. Please confirm your participation and presentation mode by replying to this email mathai.club@yandex.ru no later than March 15, 2026 18:00 Moscow time.
2. If you plan to attend in person, the organizing committee will provide accommodation details separately.
3. Please prepare your final camera-ready manuscript according to the formatting guidelines available at https://mathai.club and upload it to OpenReview by March 15, 2026 18:00 Moscow time.

Should you have any questions regarding the program, logistics, or your presentation slot, please do not hesitate to contact us.

We look forward to your contribution to MathAI 2026.

With kind regards,

MathAI 2026 Program Committee
International Conference on Mathematics of Artificial Intelligence
https://mathai.club
OpenReview: https://openreview.net/group?id=mathai.club/MathAI/2026/Conference
Telegram: https://t.me/MathAI_club
Email: mathai.club@yandex.ru